# An intercomparison of models predicting growth of Antarctic krill (*Euphausia superba*): The importance of recognizing model specificity

**Dominik Bahlburg**[1,2¤]*, **Sally E. Thorpe**[3], **Bettina Meyer**[4,5,6], **Uta Berger**[1], **Eugene J. Murphy**[3]

**1** Institute of Forest Growth and Forest Computer Sciences, Faculty of Environmental Sciences, Technische Universität Dresden, Dresden, Sachsen, Germany, **2** Department of Ecological Modelling, Helmholtz Centre for Environmental Research, Leipzig, Sachsen, Germany, **3** Ecosystems team, British Antarctic Survey, Cambridge, United Kingdom, **4** Institute for Chemistry and Biology of the Marine Environment, Carl-von-Ossietzky Universität, Oldenburg, Germany, **5** Polar Biological Oceanography, Alfred Wegener Institute for Polar and Marine Research, Bremerhaven, Germany, **6** Ecosystem Functions, Helmholtz Institute for Functional Marine Biodiversity, Oldenburg, Germany

¤ Current address: Chair of Forest Biometrics and Systems Analysis, Technical University Dresden, Tharandt, Sachsen, Germany
* dominik.bahlburg@tu-dresden.de

**Data Availability Statement:** The code for the growth models, the simulation runs and the

## Abstract

Antarctic krill (*Euphausia superba*) is a key species of the Southern Ocean, impacted by climate change and human exploitation. Understanding how these changes affect the distribution and abundance of krill is crucial for generating projections of change for Southern Ocean ecosystems. Krill growth is an important indicator of habitat suitability and a series of models have been developed and used to examine krill growth potential at different spatial and temporal scales. The available models have been developed using a range of empirical and mechanistic approaches, providing alternative perspectives and comparative analyses of the key processes influencing krill growth. Here we undertake an intercomparison of a suite of the available models to understand their sensitivities to major driving variables. This illustrates that the results are strongly determined by the model structure and technical characteristics, and the data on which they were developed and validated. Our results emphasize the importance of assessing the constraints and requirements of individual krill growth models to ensure their appropriate application. The study also demonstrates the value of the development of alternative modelling approaches to identify key processes affecting the dynamics of krill. Of critical importance for modelling the growth of krill is appropriately assessing and accounting for differences in estimates of food availability resulting from alternative methods of observation. We suggest that an intercomparison approach is particularly valuable in the development and application of models for the assessment of krill growth potential at circumpolar scales and for future projections. As another result of the intercomparison, the implementations of the models used in this study are now publicly available for future use and analyses.

generation of the figures used in this manuscript is publicly accessible at https://github.com/dbahlburg/krillGrowthModelsComparison. The chlorophyll a, sea surface temperature and photoperiod climatology-datasets used in our simulations as well as the simulation results are accessible under https://zenodo.org/record/7560809.

**Funding:** UB, BM and DB were supported by the German Research Foundation (DFG, grant number 411096565). DB was also supported by the Professor Bingel Stiftung of the German Academic Exchange Service (grant number 57646206). SET and EJM were supported by NC-ALI funding to the Ecosystems team at the British Antarctic Survey. UB was also supported by the Technical University Dresden professorship funding. BM was also supported by internal fundings of the Alfred Wegener Institute. The funders had no role in study design, data collection and analysis, decision to publish, or preparation of the manuscript. There was no additional external funding received for this study.

**Competing interests:** The authors have declared that no competing interests exist.

# Introduction

Globally, habitats are changing as a consequence of anthropogenic climate change and expanding human activities such as fisheries or agriculture with often unknown consequences for ecosystems and their functioning [1–5]. In order to mitigate and manage the impacts of these changes, it is important to understand and predict how species respond to different environmental stressors. Assessing the responses that range from the scale of individuals [6, 7] to populations [8, 9] or whole ecosystems [10, 11] requires a quantitative characterisation of these relationships. Biophysical models are a commonly used tool that allows researchers to explicitly predict the response of organisms or ecosystems to changing environmental conditions. However, a good understanding of the fundamental assumptions and limitations of the models is required to ensure their appropriate application. Therefore, there is considerable value in understanding the basis on which models were constructed and the limits of their applicability, which is essential to make reliable predictions and guide future developments.

In this study, we assess the assumptions and limitations of a suite of models that predict the growth and development of Antarctic krill under varying environmental conditions. Antarctic krill (*Euphausia superba*, hereafter krill) is a pelagic crustacean with a circumpolar distribution and arguably the most abundant species in the Southern Ocean ecosystem [12–14] with an important role in its biogeochemical cycles (e.g. [15–18]). In light of its importance, rapidly changing krill key habitats [19, 20] and a growing commercial krill fishery [21–23], there is considerable demand to understand and predict krill ecology in the past, present and future [24, 25].

Numerous krill models have been developed and applied in questions of krill ecology, natural resource management, spatial and temporal population dynamics in key habitats [26–30] and future krill habitat quality [31–34]. While these studies utilized a diverse suite of krill models, we focus on those that predict the growth of individual krill. Growth rates of individual krill directly relate environmental conditions to krill body size/mass, which is often considered a master trait for planktonic organisms [35–37]. Being able to predict krill growth and body size under varying environmental conditions is therefore key to deriving additional insights into physiological condition, population dynamics and changes of biogeochemical fluxes. However, studying krill growth is not straightforward and many different studies have led to a range of conclusions about the mechanisms of growth and and how it can be inferred from length-at-age relationships (summarized in Reiss et al. (2016) [38]). Most importantly, there are no reliable methods for determining the age of krill, although potential techniques are being investigated [39]. Further complexity is added by the apparently high rates of transfer and mixing of krill from different regions with potentially very different growth histories [40, 41]. The ability to shrink or even regress into sexually immature stages [42, 43] complicates the interpretation of length-frequency distributions and hence the identification and tracking of year-class cohorts. These difficulties, together with the large variability in environmental conditions across the Southern Ocean, make it likely that krill growth and indices such as length-at-age relationships are regionally specific [38, 44].

## Krill growth models and aims of this study

Krill models that relate growth to varying environmental conditions have been developed for over 20 years (S1 Fig). Hofmann et al. (2000) [45] introduced the first mechanistic krill growth model that was capable of predicting growth under different food concentrations. The model was initially parameterized with data from the Western Antarctic Peninsula and served as a baseline for several successors that added further functionalities such as temperature dependence, intermoult periods (IMPs) and reproductive development [46, 47]. In a separate

approach, Jager and Ravagnan (2015) [48] and later Bahlburg et al. (2021) [49] developed krill growth models that were embedded in the framework of Dynamic Energy Budget theory (DEB, [50]). Ryabov et al. (2017) [28] developed a stage-structured krill growth model based on energy partitioning and allocation to growth and metabolism that was used for reproducing population dynamics at the Western Antarctic Peninsula [51]. All the above models represent mechanistic krill growth models that explicitly consider food uptake, physiological functions and energy redistribution to predict growth. A second suite of krill growth models that were developed are empirical growth models [31, 44, 52, 53]. The models of Tarling et al. (2006) [52] and Kawaguchi et al. (2006) [53] were the first to provide parameterizations for predicting IMPs as a function of environmental conditions. They served as stepping stones for other models such as Wiedenmann et al. (2008) [31], that included moulting as a driving process of krill growth trajectories. In the Wiedenmann et al. (2008) model [31], krill remain at a constant size until they moult. On the day of moulting, the growth increment functions presented by Atkinson et al. (2006) [44] are used to predict the change in length, leading to stepwise growth trajectories and providing information on the exact days of moulting (which is important e.g. for assessing carbon flux from the shed exoskeleton, [18]). Rather than explicitly including krill growth mechanisms, the empirical models were developed by correlating observed growth rates to environmental conditions in the southwest Atlantic and Indian sectors of the Southern Ocean. The advantages of the empirical model approach are its computational efficiency, its close connection to observations and its ability to make predictions without requiring precise knowledge of the underlying ecophysiological mechanisms, within the limits of the data to which they were fitted. The main advantage of the mechanistic approach is the explicit inclusion of ecophysiological mechanisms. This arguably improves the ability to predict krill growth under environmental conditions other than those to which the models have been fitted, as the underlying mechanisms of growth are represented in the models. As identifying, characterising and parameterising the mechanisms of growth can be very challenging in practice, having the two approaches is valuable as they provide different perspectives and the opportunity for comparison of the outcomes.

What all existing krill growth models have in common is that they aim to predict growth and development of krill individuals under varying environmental conditions. However, differing design principles and the temporal and spatial context of their development make it unclear how generalizable predictions are within and between models. A key difference between the mechanistic and empirical models is that the former were designed to predict krill growth throughout the entire year whereas the empirical models were developed using datasets mainly coming from austral summer (although they included conditions further south that were considered as still being spring-like, [44, 52]). It should still be mentioned that the hypothetical full seasonal coverage of the mechanistic models ignores the considerable uncertainty that exists around the overwintering of krill, and due to the low numbers of observations available, to date, no model exists that fully captures the complexity of krill overwintering.

In our study, we assess the structures and limitations of these models, seek to understand the differences, and consider the generality between them. Generality in this context is assessed in two ways: 1. It refers to the way in which a given model is suitable for extrapolation in geographic space, in time, and in environmental space (i.e., outside the environmental conditions for which it was calibrated). 2. Generality also refers to the differences in predictions between models when simulated under the same environmental conditions. If the models evaluated in this study predict comparable results, it means that the mechanisms of krill growth (or their statistical approximations in the case of empirical models) are general and hold across a wide range of habitats and seasons, since the models originate from different regions and time periods.

We started by reviewing the relevant literature to identify all existing krill growth models that, in theory, could be used for answering similar questions about krill biology and life history under changing environmental conditions. After a pre-selection process, we implemented eight models and systematically assessed the specific data requirements of each, how temperature and food availability are interpreted and their sensitivity to these. Temperature and food availability are two key drivers of krill growth and two environmental variables that are expected to significantly change in a future Southern Ocean [54, 55]. With the above steps, we were able to characterize each krill growth model based on its inherent properties.

To illustrate the importance of understanding the specific operation of each model, we estimated krill growth rates using the same set of satellite-derived circumpolar environmental input data and compared the predicted temporal and spatial krill growth dynamics. Our analyses also revealed model mechanisms responsible for the strongest variation between models, identifying processes related to krill growth that would benefit most from an improved future understanding. Our results will contribute to the development of krill growth models that are better suited for extrapolation in environmental space, particularly under future change.

## Results

### Selecting the krill growth models

To meet our selection criteria, a krill growth model needed to be able to predict the change in length or mass of a krill individual as a function of at least one environmental input variable. This excluded the various van-Bertalanffy growth models that predict krill growth only as a function of time [56–58]. Fig 1 shows a compiled list of 17 krill growth models that met this criterion and that were further classified according to the life stage(s) they consider and the environmental input variables each required. The eight models above the horizontal line in Fig 1 are the ones that were implemented for our study. They represent a range of model types from the pioneering models of Hofmann et al. (2000) [45] and Fach et al. (2002) [46] to more recent models based on DEB theory [48, 49] and empirical models [31, 44, 52] that were closely fitted to field observations. All of these models share the use of water temperature and food availability in the form of chlorophyll concentration and/or organic carbon as drivers of growth, which allows for a meaningful intercomparison. The remaining nine models were excluded from the analysis since some of them were either conceptual [59] or too specific for life stages that the other models did not consider [60–62]. The dynamic optimization models of Alonzo and Mangel (2001) [63] and Richerson et al. (2015) [64] are backward-in-time models that optimize krill life-history decisions to maximize fitness, and thus did not fit with our research goal of evaluating models in the context of predicting krill growth potential and habitat quality under current and future environmental change. Other models require further details to allow full implementation [65, 66]. An overview of existing krill growth models and how they relate to each other is provided in Fig 1 and a table documenting the model selection can be found in the Supplementary Material (S1 Table).

### Properties of the selected krill growth models

The eight models that we implemented required a combination of water temperature, chlorophyll a concentration, particulate organic carbon (POC), photoperiod and/or day of year as environmental input variables (Fig 1).

The models of Atkinson et al. (2006) [44] and Tarling et al. (2006) [52] are empirical models that were generated by correlating observations of growth rates and IMPs with water temperature. Although not a standalone growth model, the model of Tarling et al. (2006) [52] describes IMP functions that were used to develop the instantaneous growth model in the study of

| Reference | included stages | | | input variables | | | | | | |
|---|---|---|---|---|---|---|---|---|---|---|
| | 🕷 | 🦐 | 🦐 | temp | chl a | poc | sea ice | light | adv. | bathy |
| *Atkinson et al.* (*2006*)* | | | ■ | ■ | ▨ | | | | | |
| *Tarling et al.* (*2006*)*-IMP function | | | ■ | ■ | | | | | | |
| *Wiedenmann et al.* (*2008*)* | | | ■ | ■ | ▨ | | | | | |
| *Bahlburg et al.* (*2021*) | ■ | ■ | ■ | ■ | ■ | | | ■ | | |
| *Hofmann and Lascara* (*2000*) | | ■ | ■ | | ■ | ■ | | ■ | | |
| *Fach et al.* (*2002*) | | ■ | ■ | ■ | ■ | ■ | | ■ | | |
| *Jager and Ravagnan* (*2015*) | ■ | ■ | ■ | ■ | ■ | | | | | |
| *Ryabov et al.* (*2017*) | ■ | ■ | ■ | | | ■ | | | | |
| *Alonzo and Mangel* (*2001*) | | | ■ | ■ | | | | | | |
| *Candy and Kawaguchi* (*2006*)* | | | ■ | ■ | | | | | | |
| *Constable and Kawaguchi* (*2018*) | ■ | ■ | ■ | ■ | ▨ | ■ | | | | |
| *Fach et al.* (*2008*) | | | ■ | ■ | ■ | | | | | |
| *Groeneveld et al.* (*2015*) | ■ | ■ | ■ | ■ | ■ | | | ■ | | |
| *Hofmann et al.* (*1992*) | ■ | | | ■ | ■ | | | | | |
| *Lowe et al.* (*2012*) | | ■ | | ■ | ■ | | | ■ | | |
| *Richerson et al.* (*2015*) | | | ■ | ■ | ▨ | | | | ■ | ■ |
| *Ross et al.* (*2000*)* | | ■ | | | ■ | | | | | |

**Fig 1. List of candidate krill growth models.** We used the eight models above the horizontal line for the simulations and analyses in this study. The three life-history stages are defined as *embryos and non-feeding larvae, feeding larval stages, juveniles and adults* (in that order). Input variables: temp—temperature, chl a—chlorophyll a, poc—particulate organic carbon, sea ice—sea ice presence, light, adv.—advection, bathy—bathymetry. Green tiles indicate that the chlorophyll a data used in the respective study were derived from satellite observations rather than in situ measurements or model output. Empirical models are labelled with an asterisk.

Atkinson et al. (2006) [44], which is based on data from the same cruise and region. The model of Wiedenmann et al. (2008) [31] is a composite empirical model that combines the instantaneous growth functions from Atkinson et al. (2006) [44] with the IMP model of Kawaguchi et al. (2006) [53]. In the development of the model of Wiedenmann et al. (2008) [31], food availability was reduced in such way as to make the model reproduce growth curves presented in Siegel (1987) [67]. To test an alternative parameterization to the model of Wiedenmann et al. (2008), we combined the IMP functions of Tarling et al. (2006) [52] with the growth functions of Atkinson et al. (2006) [44]. The results from these simulation runs are and labelled "Tarling et al. (2006)" in the Figures.

The two IMP models [52, 53] are structurally different in the sense that the model of Tarling et al. (2006) [52] uses body length, life stage and temperature to predict IMPs whereas the model of Kawaguchi et al. (2006) [53] only considers temperature. In addition, the model of Kawaguchi et al. (2006) [53] describes the relationship between IMPs to temperature using an exponential decay, while the shape of the model of Tarling et al. (2006) [52] is hyperbolic (S3 Fig). As a result, the IMPs predicted by both models can be very different, especially at temperatures <1˚C or >3˚C. A consequence of these structural differences is that the model of Tarling et al. (2006) [52] predicts higher moulting frequencies for most temperatures and body sizes <45 mm (we investigated the "juvenile" and "adult female" parameterization of the model of Tarling et al. (2006) [52], S3 Fig). The observations, used for the parameterization of the three empirical models, were made in the austral summer but also included more southern regions in the Scotia Sea that were still experiencing spring-like environmental conditions. While Atkinson et al. (2006) [44] and Tarling et al. (2006) [52] exclusively used data from the Atlantic sector of the Southern Ocean (South Georgia, Scotia Sea, northern Antarctic Peninsula), which is within the major region of Antarctic krill abundance [68, 69], the IMP model used by Wiedenmann et al. (2008) [31] was developed with observations from the Indian sector of the Southern Ocean ([53], Fig 2).

The five mechanistic models simulate the growth of krill according to rules of energy intake and subsequent allocation into growth metabolism and, depending on the model, additional life-history functions such as reproduction. They were all developed with the intention of covering the full life cycle of krill and therefore, in contrast to the empirical models, provide full seasonal coverage (Fig 2). With the exception of Hofmann et al. (2000) [45] and Fach et al. (2002) [46] whose models were based on experimental work on data from the Western Antarctic Peninsula, the mechanistic models incorporate observations from multiple studies at different locations. The model of Jager and Ravagnan (2015) [48] was built based on a slightly simplified version of the theoretical framework of DEB theory. DEB theory attempts to provide a unifying theoretical framework for modelling the growth and reproduction of most organisms based on shared fundamental principles. For instance, physiological processes such as metabolic rates or ingestion scale with the "volumetric length" of the krill. The volumetric length is derived from the actual body size using a shape conversion factor and refers to the body volume per unit of length. DEB theory assumes that different metabolic processes scale differently with this metric, e.g. respiration rates scale with an exponent of 3, ingestion rate with 2. In the case of the model of Jager and Ravagnan (2015) [48] and Bahlburg et al. (2021) [49], these different allometric scaling functions lead to a cessation of krill growth at a body length of $\sim$60 mm due to the increasingly unfavorable ratio of respiration rates to energy uptake at larger body sizes. The model of Jager and Ravagnan (2015) [48] was the first DEB theory model developed for krill and is partly based on phenomenological descriptions rather than measured parameter values due to a lack of data. For example, the model requires information on the fraction of energy that is allocated to growth and reproduction, a parameter that has not been clearly been defined, and that may vary in time and space. Other parameters that are highly uncertain are the mechanisms for spawning or estimating the spawning output. In the model of Jager and Ravagnan (2015) [48], krill also lack any seasonality in their physiology, which depends only on water temperature and volumetric length. This issue was addressed by the model of Bahlburg et al. (2021) [49], which included an explicit effect of the photoperiod (day length) on respiration and ingestion rates. The light-dependent scaling function was derived from experimental work by Piccolin et al. (2018) [70] but it is unclear to what extent the observed relationship is generalizable. However, it allowed, for the first time for a krill model, the simulation of an adaptive metabolism based on ambient light conditions. This was in line with the observations described by Tarling (2020) [71] that the reduction winter

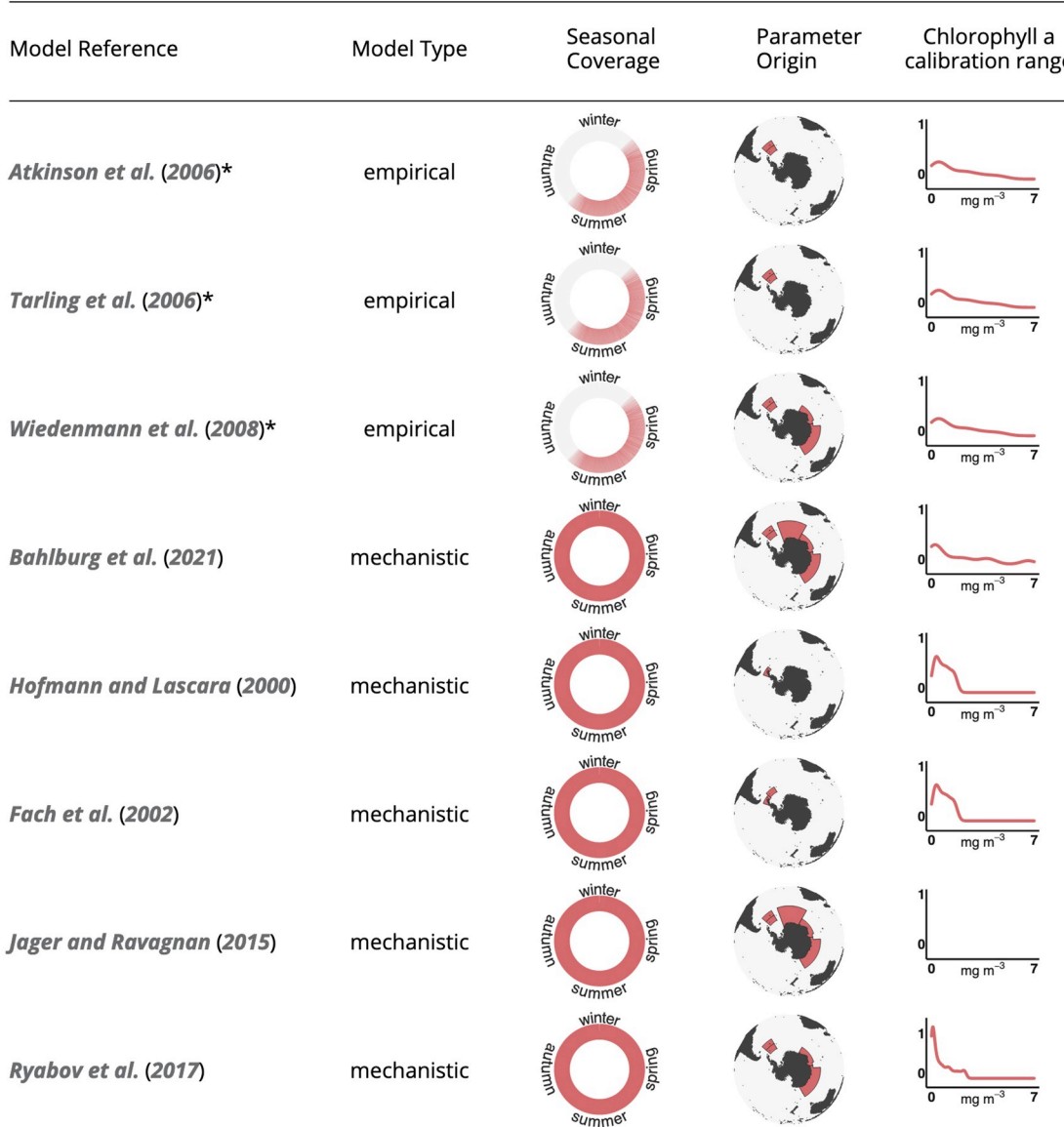

**Fig 2. Properties of the eight implemented growth models.** The chlorophyll a calibration range for Jager & Ravagnan (2015) could not be determined since in its original publication unlimited food was assumed. The calibration ranges are shown as smoothed curves that represent the relative frequency of chlorophyll a concentrations in the different calibration datasets. The three empirical models are labelled with an asterisk.

metabolism of krill is more pronounced in their high-latitude habitats and not uniform across krill habitats. In contrast, the mechanistic models of Hofmann et al. (2000) [45], Fach et al. (2002) [46] and Ryabov et al. (2017) [28] include a simple metabolic switch that reduces krill metabolism by a fixed factor for a predefined time period, typically from late fall to early spring. Although there have been attempts to include more detail from krill winter observations into growth models (e.g. Bahlburg et al. (2021) [49]), there remain considerable gaps in our knowledge of krill overwintering, mainly due to the paucity of observations and the inaccessibility of the Southern Ocean during winter. Therefore, although the mechanistic models theoretically provide full seasonal coverage, it is difficult to validate their growth- and

physiological predictions during winter. In addition to their mechanistic differences, each of the mechanistic models was fit to a specific set of environmental input data, which had different origins depending on the model. Therefore, each model was calibrated with unique ranges of water temperatures, chlorophyll a concentrations and other, environmental input variables. The models developed by Hofmann et al. (2000) [45] and Fach et al. (2002) [46] used the output of regional physical and biogeochemical models to derive environmental time series for chlorophyll a concentrations, ice algae and water temperatures. Bahlburg et al. (2021) [49] used in-situ data for temperature and chlorophyll a concentrations measured at the Western Antarctic Peninsula and Ryabov et al. (2017) [28] developed a simple sinusoidal model to approximate in-situ data of chlorophyll a concentrations off the Western Antarctic Peninsula. For its development, the model of Jager and Ravagnan (2015) [48] assumed unlimited food availability for most of its validation. We slightly modified this assumption in our simulations by scaling food availability using the Holling type II function of Bahlburg et al. (2021) [49], which is based on the same model, to include the seasonality of food availability in krill growth trajectories.

## The functional relationship of temperature, chlorophyll a and growth rates

Krill growth, to a large degree, is determined by ambient water temperatures (driving metabolic rates) and the amount of available food (driving energy assimilation). It is difficult to quantify food concentrations over large scales and chlorophyll a concentrations are a frequently used proxy of the available food. Since krill is not a strict herbivore, some models use particulate organic carbon as food proxies to account for heterotrophic food sources (Fig 1). In our simulations, we convert from chlorophyll a to particulate organic carbon by using a conversion constant of 50 following previous studies [45, 66] (further details in the Methods).

In the following, we assess the growth rates that are predicted by the eight krill growth models over gradients of chlorophyll a (as a proxy for food concentration) and water temperature. This gives us insights into the mechanistic differences in how each model relates growth to these key environmental variables. Fig 3 shows the daily growth rates predicted by each model for a 26 mm krill individual under a range of chlorophyll a concentrations and temperature during peak austral summer. The body size and timepoint were chosen to minimize the impact of other factors that influence krill growth in some of the models (e.g. reduced metabolism in winter, lipid reserves of adult krill) and the range of temperature and chlorophyll a concentrations are representative of the conditions found in the circumpolar habitats of krill.

As expected, there are clear differences in how the relationship between the two environmental drivers and krill growth is interpreted in each model. First, the maximum growth rate that each model predicts under the conditions tested here varies strongly as does the rate at which the models converge to it (Fig 3). The highest maximum growth rates for this size krill are predicted by the model using the IMP functions of Tarling et al. (2006) [52] and that of Atkinson et al. (2006) [44] with peak values of >0.3 mm d$^{-1}$ (Fig 3). The rate at which growth rates increase with increasing chlorophyll a concentrations is slower in all other models compared to the models of Atkinson et al. (2006) [44] and Tarling et al. (2006) [52]. In contrast to most other models, the model using the IMP functions of Tarling et al. (2006) [52] predicts its highest growth rates for this size of krill when temperatures are at the low end of the explored range (around -1˚C), and it generally predicts higher growth rates than the other models in most cases where temperatures are below 1.5˚C (Fig 3). In contrast, the model of Wiedenmann et al. (2008) [31] which also incorporates IMPs and uses the same growth increment function as Atkinson et al. (2006) [44], predicts its highest growth rates when temperatures are >1˚C. The model of Atkinson et al. (2006) [44] also predicts growth

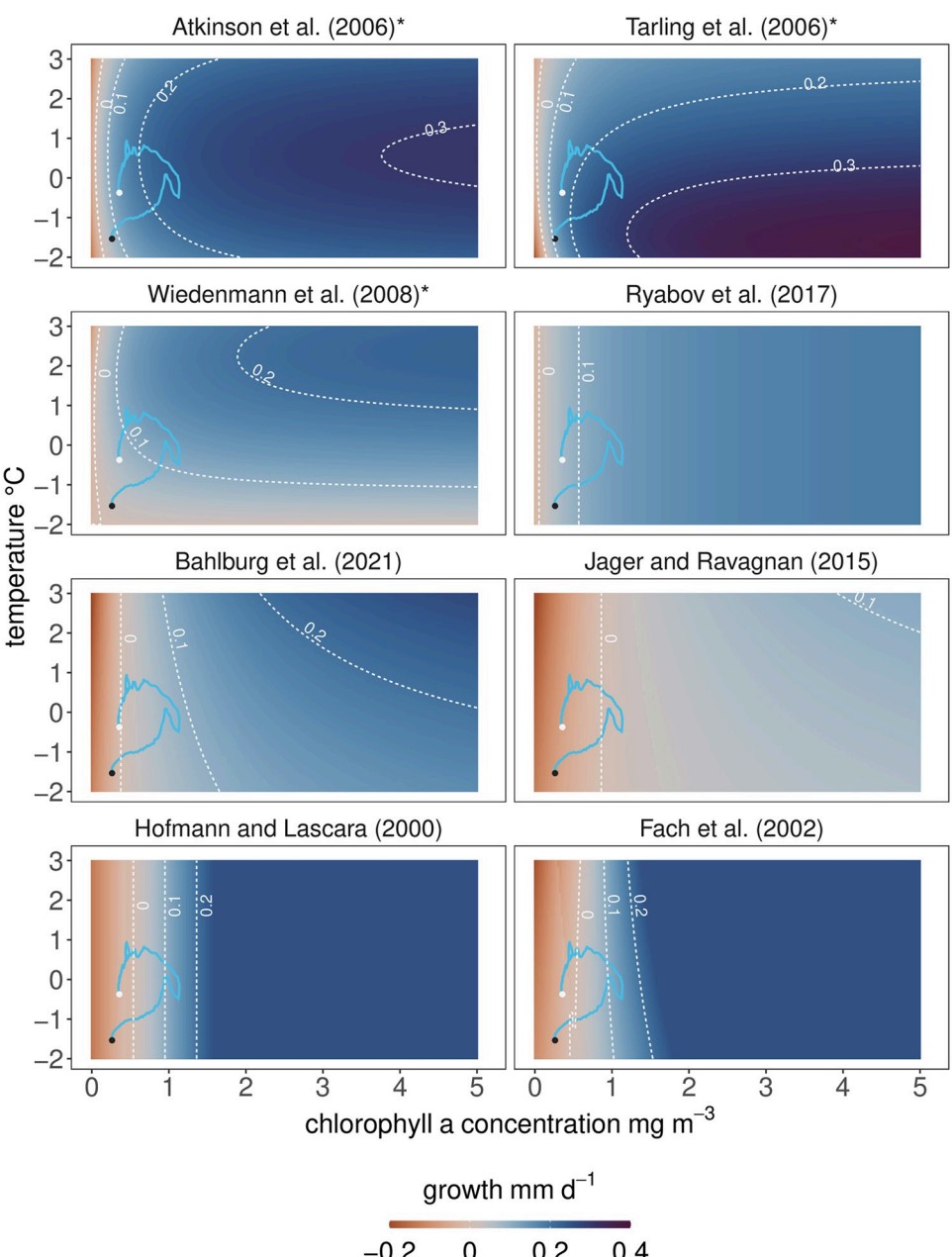

**Fig 3. Daily growth rates predicted by the models for different combinations of temperature and chlorophyll a concentrations—Two key drivers of krill growth in summer.** The growth rates are calculated for a 26 mm krill individual on 15th January (day length: 20 hours) to compare the models at the peak growth season. The blue lines show the temperature and chlorophyll-dynamics at the South Orkney Islands for the 1st November (white circle) to 15th April (black circle) to indicate the input data used in the krill growth simulations in the following section (exact location shown in Fig 4). The growth planes for the models of Atkinson et al. (2006) [44] and Tarling et al. (2006) [52] were generated using their "juvenile" parameterizations (see Methods section). The three empirical models are labelled with an asterisk.

rates of 0.2–0.3 mm d$^{-1}$ for chlorophyll a >1 mg m$^{-3}$ and temperatures -2–3°C. Of the five other models, the model of Fach et al. (2002) [46] and Hofmann and Lascara (2000) [45] predict their highest growth rates with 0.25 mm d$^{-1}$ at chlorophyll a concentrations 1.4–1.7 mg m$^{-3}$ and higher, depending on the temperature. The comparatively small effect of the

temperature dependence of metabolism on growth rates in the model of Fach et al. (2002) compared to the model of Hofmann and Lascara (2000) was already documented by Fach et al. (2002) [46]. Above chlorophyll a concentrations of 1.4–1.7 mg m$^{-3}$, growth rates in these two models are artificially capped at 0.25 mm d$^{-1}$, preventing higher growth rates that would be theoretically possible.

For most combinations of temperature and chlorophyll a, the mechanistic models predict comparatively lower growth rates than the empirical models (Fig 3) for this size of krill. For certain combinations of temperature and chlorophyll a concentration, the predicted direction of growth can be opposite depending on the model (e.g. negative growth in Jager and Ravagnan (2015) [48] vs. positive growth in Tarling et al. (2006) [52] for 1 mg chl a m$^{-3}$ and 0˚C). The areas where such qualitative differences are especially strong (ranging from negative to positive growth) are found for chlorophyll a concentrations of 0–1.5 mg m$^{-3}$. Quantitatively, these differences persist for almost all combinations of chlorophyll a and temperature but at chlorophyll a concentrations >1 mg m$^{-3}$, the models agree in predicting positive growth (Fig 3).

In summary, as expected from the model setups, there are strong differences in how the models predict growth as a function of chlorophyll a concentrations and temperature. Three key differences are

1. the minimum environmental conditions required for positive growth

2. the rate at which growth rates increase with increasing food availability and temperatures

3. the maximum growth rate returned by each model

The main reason for these differences is that the models were built and calibrated against very different food availability datasets from different locations, seasons, and numbers of observations. As a result, the empirical models [31, 44, 52] are characterized by a high sensitivity to the range of satellite derived estimates of chlorophyll a concentrations used in this study and on which the models were based. The five mechanistic models, on the other hand, are insensitive to chlorophyll a concentrations <1 mg m$^{-3}$ (visible as broad areas of low growth rates of <0.1 mm d$^{-1}$) and require higher food concentrations for more rapid growth (see S5 Fig which shows chlorophyll a concentrations required by each model to predict growth rates of 0.1 and 0.15 mm d$^{-1}$ for 26 mm krill). As a result, growth rates with increasing chlorophyll a are generally lower, as are maximum growth rates. The models of Bahlburg et al. (2021) [49] and Jager and Ravagnan (2015) [48] predict the lowest growth rates over the ranges of chlorophyll a and temperature assessed.

## Model differences in simulating growth trajectories

The growth rate patterns analyzed in Fig 3 are calculated for a particular point in time (peak summer) at a given body length (26 mm) and therefore provide only a snapshot of the potential model differences should the models be applied through the duration of the growing season. Whilst we acknowledge this is beyond the intended use of some of the models, particularly the empirical models of Atkinson et al. (2006) [44], Tarling et al. (2006) [52] and Wiedenmann et al. (2008) [31], we nevertheless apply the models for a full growing season (November to April) to illustrate the cumulative effects of the previously revealed differences. We therefore predict krill growth from November 1st to April 15th in a relatively productive area south of the South Orkney Islands in the southwest Atlantic sector of the Southern Ocean (Fig 4). The South Orkney Islands region is a key habitat, an important krill fishing ground and located within area 48.2 of the Food and Agriculture Organization of the United Nations

**Fig 4.** Top: Growth trajectories predicted by the different growth models for a location close to the South Orkney Islands—the exact location is shown in the map in the top left corner. The blue shaded area depicts the standard deviation of the outputs of all models at a given timepoint. The models strongly diverge with some models predicting very high (Tarling et al. (2006) [52]) and others predicting very low (Jager and Ravagnan (2015) [48]) growth. Bottom: Environmental data (sea surface temperature and chlorophyll a concentration) time series used to drive the growth trajectories in the top panel. The three empirical models are labelled with an asterisk. For body sizes <35 mm, the models of Atkinson et al. (2006) [44] and Tarling et al. (2006) [52] operate using the "juvenile" parameterizations for daily growth rate, growth increment and IMP, for sizes >35 mm the model of Atkinson et al. (2006) [44] uses the "all krill" parameterizations, the model of Tarling et al. (2006) [52] the "adult female" paramterization for growth increment and IMP.

(FAO), an important subunit in krill stock management [72, 73]. We parameterized the environmental conditions using satellite-derived climatologies for chlorophyll a and temperature.

The suite of growth models predicts very different growth trajectories when applied for a full season (Fig 4). Final body lengths at the end of the simulation using the model parameterisation choices detailed in the Methods section range from 19 mm (Jager and Ravagnan (2015) [48], net shrinkage of 7 mm) to 60 mm (Tarling et al. (2006) [52], net growth of 34 mm). Most models agree that in the highly productive months between December and February, growth rates are positive and the body length increases at its highest rate. The rate of growth during this time, however, differs, with the empirical models of Tarling et al. (2006) [52], Atkinson et al. (2006) [44], Wiedenmann et al. (2008) [31] and the mechanistic model of Ryabov et al. (2017) [28] predicting the highest growth rates. In periods outside of these highly productive months, model responses diverge more strongly. At the beginning of the simulation in November, with water temperatures of -1.5°C and chlorophyll a concentrations of 0.3 mg m⁻³, most of the mechanistic models predict body shrinkage whereas the empirical models and the model of Ryabov et al. (2017) [28] have positive growth rates. The first four weeks of the simulation also represent a period where the empirical models predict growth rates outside the temperature ranges of the data to which they were fitted (-1°C is the lowest temperature in the

data sets used to fit the growth functions in Atkinson et al. (2006) [44] and the IMP function in Tarling et al. (2006) [52]). Ultimately, the strong differences between models are a result of the divergence in model behaviour noted from Fig 3, when chlorophyll concentration and temperature are comparatively low. These qualitatively and quantitatively different model behaviours lead to divergence in the growth trajectories and result in a range of krill body lengths at the onset of the more productive period in December (from 23 mm in the model of Jager and Ravagnan (2015) [48] to 30 mm in the model using the IMP functions of Tarling et al. (2006) [52]).

When chlorophyll a concentrations decrease towards the end of February, all of the mechanistic models, other than that of Ryabov et al. (2017) [28], predict zero growth or shrinkage whereas the empirical models continue predicting positive growth, leading to further divergence in body length. By the end of the simulations, the three empirical models predict the highest growth, closely followed by the model of Ryabov et al. (2017) [28] which predicts a similar final body length as the model of Wiedenmann et al. (2008) [31]. On the other hand, the mechanistic models of Jager and Ravagnan (2015) [48], Bahlburg et al. (2021) [49], Hofmann et al. (2000) [45] and Fach et al. (2002) [46] predict final body lengths of 19–28 mm, meaning that net shrinkage or only little growth occurred over the simulated time period. The simulations illustrate that the differences outlined in Fig 3 accumulate and amplify model divergence when growth trajectories are projected. For completeness, in a separate run, we also simulated the model of Jager and Ravagnan (2015) [48] assuming unlimited food and only temperature regulating growth as was done in its original publication. The result is a 15 mm increase in body size for the area of the South Orkney Islands.

## Temporospatial patterns of krill growth

Some of the krill growth models require additional environmental input variables such as photoperiod or the day of year (Fig 1), which is not well reflected in the previous simulations where only one timepoint (Fig 3) or only one location (Fig 4) was considered. To demonstrate how the model differences unfold when all environmental variables are interacting on large scales, we simulated the growth of krill across the circumpolar Southern Ocean from November 1st to April 15th using climatological environmental data for all areas south of the Polar Front, which approximates the northern limit to the distribution of krill ([69], Fig 5).

As expected, based on the fixed location analysis (Fig 4), the maps depicting the predicted total change in krill length on the final day of the simulation reveal strong differences between the models. Again, the three empirical models predict the greatest change in body length. The models of Atkinson et al. (2006) [44] and Tarling et al. (2006) [52], developed to predict summer growth rates, predict length increases of over 20 mm in large areas of the Southern Ocean when used for from spring to fall. The areas of high growth align well with areas of seasonal mean chlorophyll a concentrations of >0.5 mg m$^{-3}$, spanning over large ranges of sea surface temperature (Fig 5). Only in areas with mean chlorophyll a concentrations <0.5 mg m$^{-3}$ or very high temperatures do the two models predict little growth or shrinkage.

The third empirical model, that of Wiedenmann et al. (2008) [31], also predicts high krill growth at the Antarctic Peninsula, the Scotia Sea, around South Georgia and in some areas of the Indian sector. These areas are characterized by seasonal mean temperatures of 0–2˚C and chlorophyll a concentrations of >0.8 mg m$^{-3}$ (Fig 5). Close to the Antarctic continent, where productivity is high but temperatures low, the model predicts very little growth, in contrast to the two other empirical models. This is caused by the IMP function of Kawaguchi et al. (2006) [53] used in this model, which predicts less frequent moulting at low temperatures compared to the model of Tarling et al. (2006) [52]. Since growth in the IMP-based models only occurs at

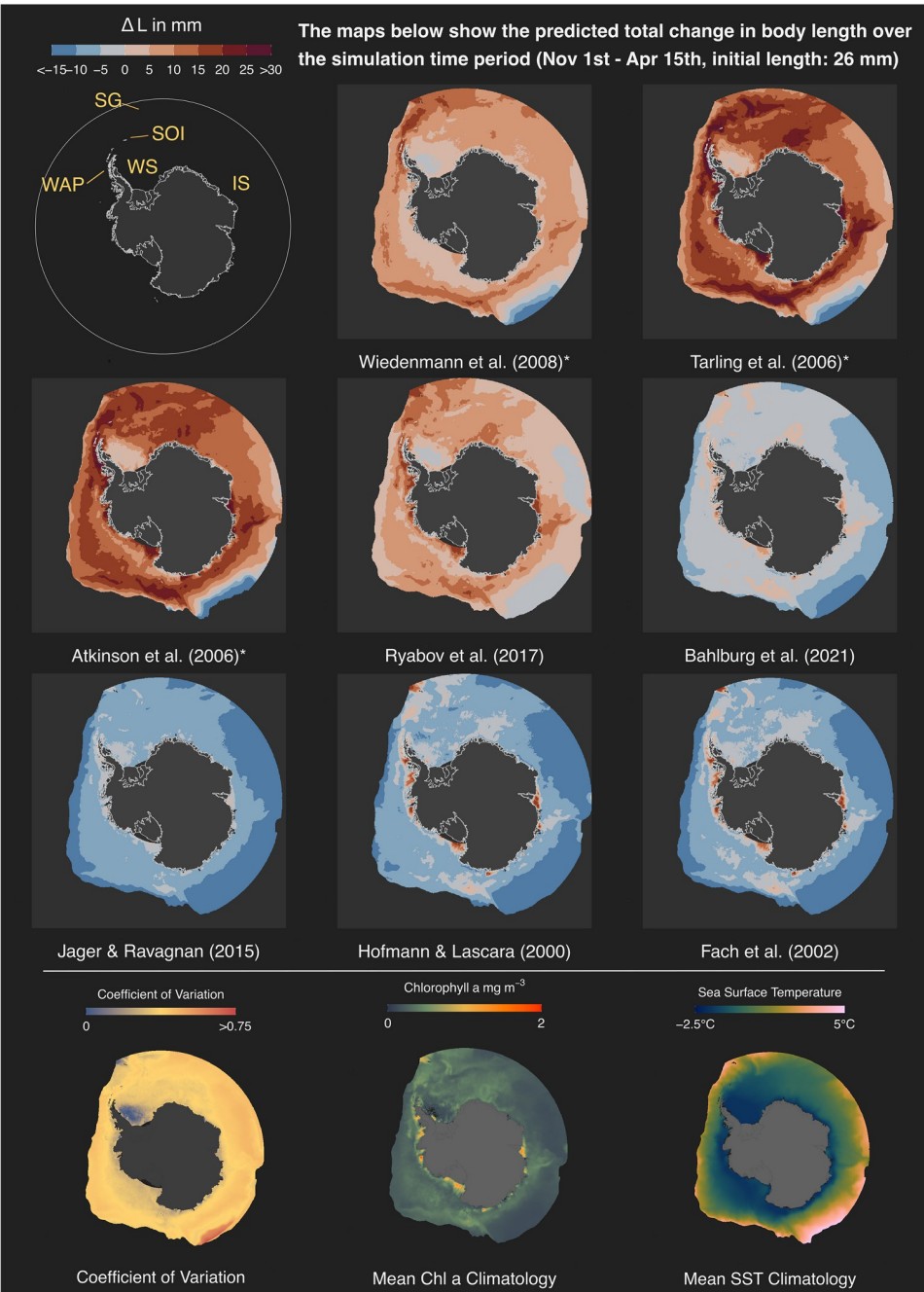

**Fig 5. Difference between final and initial krill length (mm) predicted by the different growth models after simulating growth from 1st November to 15th April for a krill of initial body length of 26 mm.** The three maps in the bottom row show the coefficient of variation of the predicted final body lengths and the mean chlorophyll a and sea surface temperature climatologies for the simulated time period, from left to right respectively. Abbreviations in the overview map: WAP: Western Antarctic Peninsula, SOI: South Orkney Islands, SG: South Georgia, WS: Weddell Sea, IS: Indian Sector. The three empirical models are labelled with an asterisk.

the day of moulting, less frequent moulting corresponds to lower overall growth when the growth increments are positive.

The simulations from the five mechanistic models, which take into account seasonal variation in metabolism, show very little net growth over the season. The mechanistic model of Ryabov et al. (2017) [28] predicts seasonal growth of $\sim 5$ mm for most regions. Restricted areas of stronger positive growth are predicted for the Scotia Sea, South Georgia, the Western Antarctic Peninsula and the highly productive but cold regions close to the Antarctic continent. However, the magnitude of maximum growth is lower compared to the empirical models and compared to the recorded growth rates on which the empirical models were based. The other four mechanistic models predict zero or negative growth for much of the Southern Ocean. The shrinkage is reduced in areas with increased chlorophyll a concentrations that correspond to the areas of highest growth predicted by Ryabov et al. (2017) [28] (Fig 3).

We use the coefficient of variation, a normalized standard deviation, to look at spatial patterns of model divergence (Fig 5). There is little spatial coherence, meaning that there is little agreement between the models across most areas of the Southern Ocean. Areas of little divergence, where the models most strongly agree, are located in the southwestern Weddell Sea and close to South Georgia. The southwestern Weddell Sea is an area characterized by almost permanent ice cover, low productivity and very cold temperatures. Considering these unfavourable environmental conditions and the assumption that growth in permanently ice-covered grid cells is zero, the models collectively predict low growth leading to a low coefficient of variation. In contrast, South Georgia is a region characterized by relatively high sea surface temperatures and high chlorophyll a concentration. In this case, most models predict positive growth of comparable magnitude leading to relatively strong model agreement. Restricted areas of very strong model divergence emerge where sea surface temperatures exceed 4˚C and chlorophyll a is in the range of 0–0.5 mg m$^{-3}$, in line with the model differences shown in Fig 3.

Similar to the comparison of growth trajectories, the conclusions drawn about the spatial distribution of productive krill habitats largely depends on the model being used and whether the model was developed to use satellite-derived data as a proxy for food availability. If the scope of this analysis would be an assessment of habitat quality, approximated by growth potential, model-specific conclusions could range from vastly unviable to circumpolar, highly productive regions. Generally, there is strong disagreement between the models in the spatial patterns predicted as suitable for positive krill growth and only extremely high or low productive regions achieve comparatively high model consensus. In regions where average chlorophyll a concentrations fall into ranges of 0–0.5 mg m$^{-3}$, which is true for the majority of analyzed areas, the predicted habitat quality can even be of opposing direction.

## Discussion

### Current limitations of krill growth models

Each of the eight models that we considered in this analysis was developed to predict growth rates of krill in line with observations under varying environmental conditions. However, our simulation experiment shows that the differences in their design, validation and calibration need to be fully understood and the models carefully applied to provide analyses of krill growth, particularly in regions and times of the year outside their initial design. Care needs to be taken to fully consider all of the assumptions noted and caveats provided in the original model studies. Simplistic application is likely to generate misleading results. However, careful application and comparison can provide valuable information for the development of more

generalized krill growth models. In a simplistic application, we observe that for a given environment, the predicted growth rates can differ significantly between models, sometimes even with opposing directions. This is not surprising but it emphasizes the limitations of the applicability of the models on large scales or with new datasets when using their default parameterizations. For practical applications such as krill habitat assessments or projection work, this means that the simulation setup as well as the input data must be compatible with the initial design of the model(s) of choice.

Compatibility of the input dataset means that it should have similar properties and methodological derivation as the original calibration data or methods for appropriate conversions need to be developed that can generate appropriate datasets. For instance, in this comparison we chose to use satellite-derived chloropyll a concentrations to provide maximum spatiotemporal coverage. Some of the growth models were defined with similar input data while others used different sources (see above). Remote sensing data are a powerful resource for parameterizing environmental conditions on large temporal and spatial scales, and frequently utilized to characterize krill habitat quality [29, 44, 46, 69, 74–76]. Nevertheless, they are also a specific data type and come with specific properties: satellite-derived chlorophyll a concentrations are not measured directly but approximated using algorithms and backscattering properties of the sea surface. It has been shown that the algorithms predicting chlorophyll a concentrations from satellite observations tend to underestimate field concentrations, especially at high latitudes and in the Southern Ocean [77–80]. In addition, remote sensing data are averaged onto spatial grid cells of fixed extents, often of tens of square kilometers. Since we used climatologies and running means from 20 years of observations, more averaging took place, effectively reducing the frequency of very high (and low) chlorophyll a concentrations. Using these input data, only four krill models were able to predict significant positive growth. Of these four, three models [31, 44, 52] make use of growth functions that were originally calibrated with satellite-derived environmental data. The fourth model, that of Ryabov et al. (2017) [28], used in-situ chlorophyll data from the Palmer Long Term Ecological Research program. The model of Bahlburg et al. (2021) [49], which struggled to predict positive growth in our simulations, also used in-situ data from a similar region but the environmental time series used by Ryabov et al. (2017) [28] included peak chlorophyll a concentrations much lower than those used by Bahlburg et al. (2021) [49]. The reason that the model of Ryabov et al. (2017) [28] is more sensitive to low chlorophyll a concentrations compared to that of Bahlburg et al. (2021) [49] can, therefore, largely be attributed to differences in the calibration data and not so much to model structure (both models use the same Holling type II food uptake function but with different parameterizations).

Additional factors for making models and simulation setups compatible are the range of simulated body sizes as well as the season and region of interest. For instance, two of the empirical models [44, 52] predict >30 mm growth during the simulation time of November to April for some regions of the Southern Ocean in our model comparison. Such growth rates are higher than those reported from other empirical growth trajectories [38, 39, 65, 81]. The effect of consistently high growth rates predicted by these models was acknowledged before by Wiedenmann et al. (2008) [31]. As a consequence, Wiedenmann et al. (2008) [31] artificially downscaled food availability to produce reduced growth trajectories. One potential reason for these high growth rates is that the empirical models do not consider the seasonal regulation of the krill metabolism—a mechanism that would effectively reduce extreme shrinkage or extreme growth outside of peak summer [49]. The seasonal regulation of metabolism is likely coupled to the photoperiod and endogenous clocks [49, 70, 81–83] and a key component of (adult) krill ecophysiology. The effects of metabolic regulation are adjusted respiration rates that vary with season and latitude due to the seasonal latitudinal dynamics of day length.

Extrapolating the empirical models to regions that differ systematically from the light- and temperature environment found across the Scotia Sea, where the variation in day length is less extreme than in areas farther south, might therefore not be appropriate. Another reason for the empirical models predicting such high growth could be that the models of Atkinson et al. (2006) [44] and Tarling et al. (2006) [52] were fitted using observations from the Scotia Sea region. This area is in the centre of the main krill distribution and a particularly good area of krill growth with very large krill being regularly observed [29, 84]. In contrast, most other models [28, 45, 46, 49] were fitted to the southern Antarctic Peninsula that represents a very different habitat.

Four of the five mechanistic models that we considered, on the other hand, struggle to predict positive growth, even in known krill habitats such as the Scotia Sea [45, 46, 48, 49, 85]. They do so primarily due to their calibration data that contain longer periods of increased chlorophyll a concentrations (e.g. Bahlburg et al. (2021) [49] contains 4 months with daily chlorophyll a concentrations >2 mg m$^{-3}$, Hofmann et al. (2000) [45] more than 3 months with >1 mg Chla m$^{-3}$). Jager and Ravagnan (2015) [48] acknowledged that their model produced daily growth rates on the lower end of observations and these are only achieved when food is unlimited, which is often not true in the field. Similarly, the Bahlburg et al. (2021) [49] model only produces growth rates comparable to the empirical models when chlorophyll a concentrations reach values of more than 4 mg m$^{-3}$ and at high temperatures (>1.5˚C). The models of Hofmann et al. (2000) [45], Fach et al. (2002) [46] and Bahlburg et al. (2021) [49] are also theoretically capable of producing high growth rates but food concentrations required for this are rarely reached in our simulations. Despite these common features, we still see a very large >20 mm difference in estimated cumulative growth rates within the mechanistic models set. While some models generate negative rates, others predict more reasonable rates for areas where high growth rates are expected, such as in the Scotia Sea when comparing the model of Jager and Ravagnan (2015) [48] with the model of Ryabov et al. (2017) [28] or Bahlburg et al. (2021) [49]. Another interesting result is that the model of Bahlburg et al. (2021) [49] predicts similar growth as the models of Hofmann and Lascara (2000) [45] and Fach et al. (2002) [46], despite being closely related to the model of Jager and Ravagnan (2015) [48]. The addition of photoperiod and temperature (using empirical data) as drivers of krill metabolism strongly reduces the energetic requirements of krill individuals in the model of Bahlburg et al. (2021) [49] compared to the model of Jager and Ravagnan (2015) [48], leading to a lower impact of spring and fall starvation periods, particularly in high-latitude habitats, and allows for more efficient ingestion of food during the summer. Our results highlight that there are some fundamental differences in the mechanistic models. Fitting the models with datasets containing longer periods of increased food availability results in parameterizations that suppress growth for chlorophyll a concentrations below 0.5 mg Chla m$^{-3}$ to keep predicted growth trajectories in line with those to which the models were fitted. In addition, much of the mechanistic model development and validation has been undertaken in more seasonal low growth habitats. Adjusting these existing models to areas where krill growth is known to be high might be a useful future step. Currently, the characteristics of input data and context of model development emphasize the clear limitations of how the existing mechanistic growth models can be used in their default parameterizations.

In summary, our simulations show that the eight investigated krill growth models do not represent a suite of general models but ones that are specific to certain regions and seasons, and constrained to technical characteristics of the datasets they were calibrated with. They are useful tools for exploring the interactions of krill with its environment when used under full consideration of their limitations.

## Towards more general krill growth models

A first step towards more general krill growth models that are applicable on large spatial scales would be a greater and more detailed cross-comparison. Data used in such cross-comparison should contain observations from multiple regions, seasons, and life stages with complementary environmental data coming from the same data source to be as representative as possible for the Southern Ocean. When using re-calibrated models, it would be important to consider the uncertainties around parameter estimates to generate confidence intervals that represent the natural variation and uncertainty in the observations. For example, Murphy et al. (2017) [29] considered uncertainty estimates in their application of the empirical growth model of Atkinson et al. (2006) [44] and highlighted the sensitivity of the estimated growth rates to variation in the temperature and food availability data. Such effort could potentially lead to three outcomes:

1. region-specific parameterizations of single models (single model, regionally adaptive parameters)

2. region-specific models, whose predictions are combined for large-scale assessments (complementary approach)

3. models (individual models or ensemble) that predict the mean response of krill in large-scale assessments, incorporating the regional variability of observations as variance around the mean

Future studies to evaluate these approaches will be an important next step in this research. Although a re-calibration of the models would reduce their specificity, some significant uncertainties remain. For instance, the functional relationship of temperature, chlorophyll a concentrations and growth is, besides quantitative differences, also structurally different between the models. While most models agree that growth rates should peak at high food concentrations (chlorophyll a >1.5 mg m$^{-3}$) and water temperatures >0˚C, the shape of these functions and the exact location of the maxima differ. Despite the impact of differing calibration data, there is not a consensus on the form of the relationship, which may be regionally specific, opening opportunities for meta-analyses and additional experiments to clarify these important questions. For example, a recent long-term experiment (8 months) investigating the effects of increasing temperatures on krill growth failed to find any effect of temperature on respiration and growth rates of adult krill in the 0.5–3˚C temperature range [86]. As ectotherms, and based on temperature-dependent enzyme kinetics, it is surprising that no long-term temperature effect on krill growth could be detected within this range. It is possible that compensatory mechanisms counteract the physico-chemical effects of elevated temperatures on molecular processes. Although, these results stand in contrast to field observations (e.g. [44, 71]), they highlight the need for long-term studies to investigate the effects of changing environmental conditions on krill growth, as most short-term experiments conducted over a few days have concluded significant temperature effects on krill metabolism, which were later used for model parameterizations [87, 88].

Another example of uncertain mechanisms are the two IMP-based models [31, 52] that predict growth as an increment in body length at the event of moulting. For the model dynamics, more frequent moulting implies higher growth rates whereas fewer moulting events correspond to less growth, when growth rates are positive. When growth rates are negative (e.g. in winter, which we did not address in this study), more frequent moulting would result in rapid shrinkage. The frequency of moulting in the two models is based on two different functions that relate the IMP to body length (taken from Kawaguchi et al. (2006) [53]) and to water temperature and body length (Tarling et al. (2006) [52]). The two IMP models differ considerably

in the range of values each predicts and the relationship between IMP and temperature. For example, predicted IMPs are predicted to increase at temperatures >0.5˚C by the model of Tarling et al. (2006) [52] whereas they decrease in the model of Kawaguchi et al. (2006) [53] (using the "adult female" parameterization of the model of Tarling et al. (2006) [52], S3 Fig; for "juveniles", the temperature-IMP relationship is slightly differently shaped with a maximum at 3˚C and lower values at lower and higher temperatures, respectively). For a temperature of 0˚C and a krill individual of 35 mm, the IMP predicted by the model of Kawaguchi et al. (2006) [53] is almost 3 times longer for a mature female krill than that of Tarling et al. (2006) [52] (30 days vs. 10 days). The strong differences between the two existing krill IMP functions indicate large uncertainty about the relationship of these variables, introducing considerable variation into predicted growth trajectories. This uncertainty was recognized in the parameterization of the IMP model of Tarling et al. (2006) [52], who found, for example, that IMPs for juveniles were little related to body length. Arguably, this within-model uncertainty is also responsible for the variability in the stage-specific functional relationships between IMPs, temperature and body size (S1 File). Therefore, additional observations of IMPs under different environmental conditions are needed to better understand this relationship and to reduce the uncertainty in the models.

Finally, our simulation setup likely underestimates growth rates predicted by candidate models that require particulate organic carbon concentrations (POC) as a food input [45, 46]. We derived POC-concentrations for these models by using a chlorophyll a:carbon conversion factor of 50 mg POC (mg Chla)$^{-1}$ as was done in their original development [45]. Using a fixed constant to convert from chlorophyll a to POC is a broad simplification as this ratio varies with phytoplankton community composition and season [89–91]. Thomalla et al. (2017) [91] showed that in the Southern Ocean, this ratio can change seasonally from values of 20 to 100 mg POC (mg Chla)$^{-1}$ with peak values in austral summer. In their observations, 50 mg POC (mg Chla)$^{-1}$ only corresponded to a seasonal mean, meaning that we potentially under- or overestimate true carbon availability depending on the time of the year and phytoplankton community composition. Addressing the dynamics of the chlorophyll a:POC ratio would allow for more accurate estimates of true autotrophic food availability. In addition, it has been shown that krill can utilize several food sources and phytoplankton (for which chlorophyll a acts as a proxy) only represent one component of krill diet, with observations of krill feeding on pelagic phytoplankton, heterotrophic organisms, sea ice biota and detritus [47, 92–95]. The relative contribution of each food source to the diet of krill is not fully resolved yet and might depend on region, season and life stage [96–98]. Typically, it is assumed that heterotrophic food and sea ice biota are of higher importance in periods when pelagic phytoplankton is scarce (e.g. in austral winter, [47, 95]). We tried to minimize the potential importance of other food sources by excluding winter from our simulations but are aware that additional food sources could have allowed for higher growth rates in the POC-based models.

Other aspects of krill feeding ecology that are important for growth and development, such as taxonomic prey composition and size preferences [93, 99] and the importance of prey lipid content and lipid composition ([76, 100, 101]), are not captured by any of the models considered, nor by common food proxies such as chlorophyll a or POC concentrations. For example, it has been shown that early access to crucial fatty acids, often contained in larger diatoms during spring phytoplankton blooms, is key for successful gonad development and spawning in adult females [100, 102, 103]. These important details are not only missing in most of the current models (some, but not all, are included in the model of Fach et al. (2008) [104]), but also in the large-scale environmental datasets required for krill habitat assessment. Significant advances in empirical studies, ocean observations, krill model development, and biogeochemical modeling are needed to address these issues.

On a shorter time scale, we propose a re-calibration and uncertainty estimation of the existing krill growth models using a comprehensive cross-comparison to investigate whether more general krill growth models can be applied on large scales with more confidence. This includes a compilation of information on size and growth rates under different situations using different techniques. These data could be used to assess how krill growth varies and also to provide the best available understanding of the various processes involved—e.g. overwintering and how this varies in different conditions. When applying models, special attention should also be paid to uncertainty estimates around parameter values as they represent natural observational variation that should be considered in the main analysis. However, additional experiments and meta-analyses of published datasets are required to resolve additional uncertainties in the mechanisms of growth and food availability.

## Conclusion

Our simulations demonstrate that the currently existing krill models that we considered are specific to their type of calibration data. When being used with a different type of input data to their calibration data, many of these models struggle to predict realistic growth trajectories, even in those regions and during the seasons that they were parameterized for. Our findings emphasize that the available growth models for krill should be used with clear consideration for their constraints, in appropriate spatial and temporal contexts and only with appropriate sets of input data that are comparable to those used for model parameterization. When the models are intended to be used with input data that systematically deviate from their calibration data (e.g. satellite data vs. in-situ data), they need to be re-calibrated for the respective data type. There is no best or worst model, but rather the regional and temporal context of a simulation study, as well as the technical specifics of the input data, largely determine the most appropriate model candidate. Nevertheless, it also becomes clear that uncertainty in the relationships underlying fundamental model building blocks, such as between temperature, food availability and growth rates leads to large uncertainty in model simulations. As with the use of multiple models to predict future climate scenarios, simulating the response of krill to environmental change by using different models in parallel could be used as an opportunity to learn more about our understanding of how the system works, and ultimately lead to more reliable predictions, not in spite of, but because of the explicit consideration of model uncertainties. We hope that our model library and open access code can contribute to this journey.

It is now commonplace to make model source code publicly available to increase transparency and reduce the risk of introducing errors when re-using models. We encourage researchers to publish source code when new models are being presented so that future krill modelling projects and intercomparison studies will be more straightforward. In addition, documenting the models in standardized ways, for instance by using the Overview, Design concepts, Detail protocol (ODD) developed for individual-based models [105, 106], not only improves transparency and reproducibility but also contributes to a more open and inclusive research culture.

## Materials and methods

### Model implementation

The eight krill growth models used in this study were implemented using the R programming language [107]. We also implemented the krill growth model of Constable and Kawaguchi (2018) [66] but were not able to accurately validate our implementation (see model quality control in the S1 File). Therefore, we excluded it for the analyses in this study but included it in our code repository.

We used the standard-parameterizations for each model as reported in their original publication. Uncertainties of parameter estimates are provided for some of the models but were not included in our analysis because we wanted to focus on the variability between-models. However, and as one important conclusion of our study, within model uncertainty should explicitly be considered in future applications. After its implementation, we validated our model version against the original published form, using the same environmental input data and/or reported model output. We extracted the environmental data from the corresponding figures in the original studies in which each model was presented, using the data extraction tool from ImageJ [108]. The resulting growth trajectories were compared with those reported in the respective paper and the model was considered adequately implemented when both matched. The complete documentation of the model quality controls can be found in the Appendix (S1 File). To run the simulations, process and visualize data, we used the following R-packages: *sf* [109], the *tidyverse* [110], *scico* [111], *terra* [112], *tidyterra* [113] and *suncalc* [114].

## Environmental input data used for the simulations

We used daily climatologies for sea surface temperatures and chlorophyll a concentrations to simulate the growth of krill using the eight different models.

To determine the sea surface climatologies, we downloaded daily sea surface temperature observations from the Copernicus Marine Service using the Global Ocean OSTIA Sea Surface Temperature and Sea Ice Reprocessed product (https://doi.org/10.48670/moi-00168) from 2003-01-01 to 2020-12-31 [115]. The dataset is based on satellite observations that were reprocessed to fill any data gaps. We extracted the data for areas south of 53˚S and downsampled them to a spatial resolution of 0.25x0.25˚ before calculating the mean sea surface temperature for each day of the year.

For the daily chlorophyll a climatologies, we used the Global Ocean Colour (Copernicus-GlobColour) L4-processed product [116], which is also publicly available from the Copernicus Marine Service (doi: https://doi.org/10.48670/moi-00281). The dataset combines global ocean colour observations from multiple sensors including SeaWiFS, MODIS and MERIS to derive sea surface chlorophyll a concentrations. The data were reprocessed and interpolated to fill gaps such as those caused by cloud cover (the full processing and a quality assessment of the dataset are documented at https://doi.org/10.48670/moi-00281). The full dataset provides data for various geochemical variables but we only used the estimated chlorophyll a concentrations in a daily resolution from 2003-01-01 to 2020-12-31. Analogous to the sea surface temperature climatologies, we downsampled the data to a spatial resolution of 0.25x0.25˚ and calculated mean chlorophyll a concentrations for each day of the year. Strong sea ice cover, especially during the austral spring, resulted in extensive data gaps. The climatologies for grid cells that were ice-covered in some years at a given day were calculated based on the available observations, effectively reducing the number of samples used for the calculating the mean. Since there were still significant data gaps, we calculated a 15-day rolling mean of the climatologies. This reduced the number of ocean grid cells without information from e.g. 50% to 25% for the earliest time points in our simulation. Regions where climatologies were based on a low number of samples (e.g. less than 4) occurred predominantly in November. Chlorophyll a concentrations in these regions were usually <0.3 mg m$^{-3}$, meaning that the predicted growth was low, consistent with the overall growth patterns at this time of year. The slight biases introduced by this applied equally to all growth models, which still allowed for a meaningful intercomparison, which was the main purpose of this study.

As a third required environmental input variable, we also calculated the photoperiod for each grid cell at each day of the year over the simulation time using the R-package *suncalc* [114]. We defined photoperiod as the hours that passed between sunrise and sunset.

Finally, we limited our simulations to areas south of the Polar Front, which marks the approximate northern limit of krill habitat [69]. The mean location of the Polar Front was calculated by averaging the weekly locations of the Polar Front from 2002 to 2014 ([117], S6 Fig).

## Simulation setup and assumptions

The circumpolar simulations of krill growth were performed on a spatial grid with a 0.25x0.25° resolution, resulting in >105 000 ocean grid cells. The simulated time period ranged from November 1st to April 15th (166 days) to span the main Southern Ocean productive period from spring to fall. This was done due to the lack of satellite observations during the austral winter and since some models were not developed to simulate growth during winter (Fig 2). Due to the missing chlorophyll a-concentrations in continuously ice-covered grid cells and the non-availability of some environmental variables such as heterotrophic carbon (required by the model of Fach et al. (2002) [46]), we made the following assumptions:

- heterotrophic carbon and ice algal concentration was set to zero since no datasets exist that plausibly estimate these environmental variables on the required spatial and temporal scales (affecting the models of Ryabov et al. (2017) [28], Hofmann et al. (2000) [45] and Fach et al. (2002) [46]). This reduces the amount of available energy in these models but, according to our current knowledge of krill ecology, these food sources are considered particularly important during the austral winter which was omitted from our simulations.

- when a grid cell is covered with sea ice at a given timepoint, growth for that day is set to zero due to the lack of under-ice chlorophyll a observations. This assumption is in agreement with observations of very low chlorophyll a concentrations in extensively ice-covered waters. For the models that are based on IMPs this meant that the IMP was increased by 1 day for each day that the respective grid cell was ice-covered.

At the start of the simulation, each grid cell was populated with one krill individual of 26 mm. The initial size was chosen as it represents the lower end of the krill size calibration range of Tarling et al. (2006) [52], and therefore, the smallest common krill size that all models were parameterized for. We then simulated the krill growth at each location in a temporal resolution of 1 day until April 15th. The time step of 1 day ensured sufficient precision in the numerical integration of the models and matched the temporal resolution of the environmental datasets. Structurally simple models, such as Atkinson et al. (2006) [44] or that using the IMP functions of Tarling et al. (2006) [52] were solved using the Euler method to reduce computation time. Some structurally complex models [28, 45, 46] were solved using the 4th-order Runge-Kutta integration method. We also tested the 4th-order Runge-Kutta integration method for the models that we eventually solved using Euler's method but the results were not sensitive to the choice of integration scheme.

The growth functions of Atkinson et al. (2006) [44] and the IMP functions of Tarling et al. (2006) [52] are available with different parameterizations for juvenile, adult male and female krill. In addition, the model of Atkinson et al. (2006) [44] provides an "all krill" version which has typically been the choice in previous applications of their growth model [29, 32, 33], and which we also used in our simulations for krill >35 mm. We considered krill <35 mm as juveniles and simulated growth for such individuals using the "juvenile"-parameterization of the model of Atkinson et al. (2006) [44]. The same was done for the model of Tarling et al. (2006) [52]. However, since the IMP function of the model of Tarling et al. (2006) [52] does not exist

in an "all krill" version, we modeled growth for krill >35 mm using the "adult female" parameterizations for the IMP model as well as the growth increment function from Atkinson et al. (2006) [44]. We tested the sensitivity of the simulation results to alternative parameterizations of these functions ("adult male" or "adult female" for the model of Atkinson et al. (2006) [44] and "adult male" or "all krill" for the growth increment function in the model of Tarling et al. (2006) [52]) but did not generate qualitatively different results (S4 Fig).

In Fig 3, we compare daily growth rates predicted by each model for various combinations of water temperature and chlorophyll a concentrations. The models based on IMPs [31, 52] do not directly return daily growth rates but consider growth as periods of constant body size (the IMP) followed by a change in body length at the point of moulting. For those models, we calculated daily growth rate as the growth increment on the day of moulting (in mm) divided by the duration of the IMP (in days). To predict the growth increment at moulting, the growth function used in both models requires information on water temperature and chlorophyll a concentration. In our simulations, we averaged the chlorophyll a concentrations and water temperatures during each IMP and used these averages to predict the growth increment on the day of moulting using the growth rates predicted by the model of Atkinson et al. (2006) [44]. In this way, growth was not only determined by the environmental conditions during moulting but also by the environmental conditions experienced during the IMP.

## Supporting information

**S1 Fig. A chronology of the development of krill growth models and their interrelationships.** IMP—intermoult period, DEB—Dynamic Energy Budget, IGR—Instantaneous Growth Rate, DVM—Diel Vertical Migration.
(TIF)

**S2 Fig. Daily growth rates predicted by the models for different combinations of temperature and chlorophyll a concentrations but using a body size of 50 mm (using the "all krill" parameterization for the model of Atkinson et al. (2006) [44] and the "adult female" parameterization for the model of Tarling et al. (2006) [52]).** Blue lines as per Fig 5.
(TIF)

**S3 Fig. Differences of the IMP-functions of Tarling et al. (2006) [52] ("juvenile" and "adult female"-parameterization for krill of different body lengths) and Kawaguchi et al. (2006) [53] (used in Wiedenmann et al. (2008) [31]).**
(TIF)

**S4 Fig. Additional simulation results showing the sensitivity of the models of Atkinson et al. (2006) [44] and Tarling et al. (2006) [52] to different stage-specific parameterizations (female krill = "mature female" in the model of Tarling et al. (2006) [52]).** Note that for krill <35 mm, the models always operate in their "juvenile" parameterization.
(TIF)

**S5 Fig. Chlorophyll a required by each model to predict growth rates of 0.1 and 0.15 mm d$^{-1}$ for a 26 mm individual in summer and a water temperature of 1˚C.** Empirical models are labelled with an asterisk.
(TIF)

**S6 Fig. Location of Polar Front derived from Freeman and Lovenduski (2016) [117] that is used as a northern boundary for the simulation results.** The black lines show the weekly locations of the Polar Front from 2002 to 2014, and the orange line the mean used to define

the northern boundary of the model simulations.
(TIF)

**S1 File. Checking the correctness of the implementation of the krill growth models.**
(PDF)

**S1 Table. Model selection process.**
(XLSX)

## Acknowledgments

We thank Eileen Hofmann and Alexey Ryabov for their support during the implementation of their krill growth models. We would also like to thank Kim Bernard and an anonymous reviewer for their careful reading and very helpful comments during the revision process.

## Author Contributions

**Conceptualization:** Dominik Bahlburg, Sally E. Thorpe, Uta Berger, Eugene J. Murphy.

**Formal analysis:** Dominik Bahlburg.

**Funding acquisition:** Bettina Meyer, Uta Berger.

**Investigation:** Dominik Bahlburg.

**Methodology:** Dominik Bahlburg, Sally E. Thorpe, Eugene J. Murphy.

**Resources:** Uta Berger.

**Software:** Dominik Bahlburg.

**Supervision:** Sally E. Thorpe, Bettina Meyer, Uta Berger, Eugene J. Murphy.

**Validation:** Dominik Bahlburg, Eugene J. Murphy.

**Visualization:** Dominik Bahlburg.

**Writing – original draft:** Dominik Bahlburg.

**Writing – review & editing:** Dominik Bahlburg, Sally E. Thorpe, Bettina Meyer, Uta Berger, Eugene J. Murphy.

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
