## [Decision Letter · Decision Letter 0]

14 Jun 2023

PONE-D-23-06152Intercomparison of growth models for Antarctic krill (Euphausia superba): towards a generalised understandingPLOS ONE

Dear Dr. Bahlburg,

Thank you for submitting your manuscript to PLOS ONE. After careful consideration, we feel that it has merit but does not fully meet PLOS ONE’s publication criteria as it currently stands. Therefore, we invite you to submit a revised version of the manuscript that addresses the points raised during the review process.

We look forward to receiving your revised manuscript.

Kind regards,

Peng Chen, Ph.D.

Academic Editor

PLOS ONE

Journal Requirements:

‘UB, BM and DB were supported by the German Research Foundation (DFG, grant number 411096565, https://gepris.dfg.de/gepris/projekt/411096565?language=en)

DB was also supported by the Professor Bingel Stiftung of the German Academic Exchange Service (DAAD, https://www.deutsche-stiftungstrust.de/dr-bingel-stiftung/) .

SET and EJM were supported by NC-ALI funding to the Ecosystems team at the British Antarctic Survey (https://www.ukri.org/councils/nerc/guidance-for-applicants/types-of-funding-we-offer/national-capability-funding/).

UB was also supported by the Technical University Dresden (professorship funding, https://fis.tu-dresden.de/portal/en/researchers/uta-berger(94fcb077-5b3b-4beb-a905-64976d5ce4b2).html)

BM was also supported by internal fundings of the Alfred Wegener Institute (https://www.awi.de/ueber-uns/organisation/mitarbeiter/detailseite/bettina-meyer.html)”

Reviewers' comments:

Reviewer's Responses to Questions

**Comments to the Author**

1. Is the manuscript technically sound, and do the data support the conclusions?

Reviewer #1: Yes

Reviewer #2: Yes

2. Has the statistical analysis been performed appropriately and rigorously? 

Reviewer #1: Yes

Reviewer #2: Yes

3. Have the authors made all data underlying the findings in their manuscript fully available?

Reviewer #1: Yes

Reviewer #2: Yes

4. Is the manuscript presented in an intelligible fashion and written in standard English?

Reviewer #1: Yes

Reviewer #2: Yes

5. Review Comments to the Author

Reviewer #1: The manuscript by Bahlberg et al. provides, for the first time, a detailed comparison of Antarctic krill growth models. This is an important study with valuable findings. It is well written, and the analytical approach has been performed at a high technical standard. The details provided in the manuscript are clear and thorough. Model code has been made available to the public for transparency. Overall, I recommend this manuscript for publication with a few minor revisions that I will outline below.

In general, I think the following concepts are not explored in the paper and I recommend that they be brought into the Discussion section somewhere:

1. Not all chlorophyll-a is accessible or preferential to krill. The only real discussion of chlorophyll-a is about total concentrations, but I think it would be pertinent to also talk about the fact that generally under high chlorophyll-a concentrations, diatoms typically dominate whereas smaller phytoplankton often dominate at lower chlorophyll-a concentrations. The latter tends to shift the food web and increase the number of trophic steps before krill – they tend to become more carnivorous. This leads into the next topic that I think the paper would benefit from a brief discussion of.

2. Not all food is equal in terms of growth. We know from previous lipid studies that krill need diatoms and some protozoans for growth. If krill aren’t eating diatoms, but are still getting a lot of food through carnivory, they might not be growing as fast or as much.

3. I was anticipating the manuscript mentioning which of the models seemed to be the most accurate. This doesn’t happen in the paper though and I think it would be helpful to have some discussion about whether there is a “best” model. Are the mechanistic models better than the empirical models or the other way around? It feels a bit like you have to read between the lines to decipher this. If it is not yet possible to say which is better, then I think this should be stated so that the reader isn’t left wondering.

Title: After reading the manuscript, I came to the conclusion that the title might be a little misleading. In particular, the part “towards a generalised understanding” suggested to me that there would be a generalised understanding of Antarctic krill growth, but I’m not convinced that is necessarily achieved here, nor am I sure that was the goal. It seems to me that the goal of the manuscript was to compare the various modeling approaches and make recommendations on how best to proceed in the future. I think the paper does this well. I would recommend slightly changing the title to something along the lines of “Intercomparison of growth models for Antarctic krill (Euphausia superba): towards a generalised approach” or “…towards a common approach” or something like that, which more clearly articulates the outcome of the paper.

Line 19: “…the most dominant…” – The word “dominant” has different connotations and “abundant” would be a better, more precise term here.

Line 28: I would rephrase to “Growth rates of individual krill…”

Line 29: Add a comma before “which”

Line 124: Delete “well” from “…were not well compatible…”

Lines 138-139: Correct sentence to “…was reduced in such a way as to make…”

Line 186: Add a comma before “which”

Line 187: Add a comma before “which”

Line 189: Because of the way it is written, it would be helpful to include the author name in the citation of “[68]” – i.e., “…from experimental work by Piccolin et al. [68]”. I noticed that this was sometimes done and other times not. I will include notes on the ones I caught, but I don’t think I noted it down in every instance so please go through the entire manuscript to make sure where you refer to a paper within a sentence as in the above example, you include the author’s name.

Line 192: This is another example of where I think you could include the author’s name – “…described by Tarling [69]” – also note that it should be “by” rather than “in”.

Line 204: Add a comma before “which”

Line 238: Change to “growth to these key environmental variables”

Lines 282-283: “operation” is used twice in the sentence. You can delete the second occurrence.

Lines 292-293: “change” is repeated twice in this sentence. You can delete the second occurrence. Also, this sentence is still quite hard to read/follow. Perhaps consider rephrasing for further clarity.

Line 356: Add a comma after “expected”

Line 371: Add a comma before “which”

Lines 373-374: This statement is made several times in the paper, but it’s not explained whether this is an incorrect assumption or not. I think it would be helpful to the reader if this was expanded on.

Lines 511-512: Change to “…specific to certain regions and seasons, and constrained to…”

Line 517: Remove the space between “-“ and “comparison in “cross- comparison”

Line 525: Add the author name, i.e., “For example, Murphy et al. [29] considered…”

Line 552: Change to “Another example of…”

Lines 554-555: I think this statement is an oversimplification. More frequent moulting could also occur if the krill need to shrink rapidly, which would not lead to higher growth rates. However, more frequent moulting and a positive growth increment will result in higher growth rates.

Line 561: I am not sure what you mean by “reversely interpreted”, please rephrase this for clarity.

Lines 589-593: See my second commend under the broad comments before the line-by-line comments. I think some discussion on the role of different food types in promoting and supporting growth is needed.

Line 624: Please write ODD out in full here.

Lines 640-654: There is no mention on the effects of clouds for ocean colour and this should probably also be included, even just briefly.

Line 685: How did you choose 1 day?

Line 730: Include author name, as in “…model of Constable et al. [64]…”

Lines 740-742: What are the “respective Figures”? Please rephrase for clarity.

Figure Captions:

Figure 3. caption on page 7: I would add “(black circle) after “15th April” to be consistent with the mention of “(white circle)” following “1st November”.

Figure 4. caption on page 9: There is a floating, unused “(“ on the fifth-to-last line.

Figure 5. caption on page 10: Remove the “.” after “1st” and before “November”

Reviewer #2: This paper compares an ensemble of empirical and mechanistic krill growth models, and assesses them based on their generality – i.e. their suitability for extrapolation in space and time. The authors conclude that the model results are sensitive to the data to which the model was calibrated, and advocate for these strengths and limitations to be accounted for in future studies applying these models, as well as the value of model re-calibration and code transparency to allow for reproducibility.

I see this study as a very valuable addition to the field of krill habitat modelling and thank the authors for their work. Firstly, krill growth models are often used to predict the impacts of climate change on krill habitat (e.g. Hill et al. 2013, Veytia et al. 2020, Sylvester et al. 2021) and an inter-comparison of growth models is valuable for assessing strengths and biases within these applications. Secondly, the availability of these models within open-source code is a valuable resource in and of itself that will assist in future reproducibility. Lastly, the authors have highlighted knowledge gaps that are important for reducing biases in large-scale and future assessments of krill habitat.

Overall, the paper and its accompanying code are well-structured and well written. I have made comments and added questions throughout the main text and supplementary files to improve clarity and flow of argument as well as to provide additional rationale/discussion. I therefore recommend publication with these moderate adjustments (see comments in attached files).

6. PLOS authors have the option to publish the peer review history of their article (what does this mean?). If published, this will include your full peer review and any attached files.

Reviewer #1: **Yes: **Kim Bernard

Reviewer #2: No

---

## [Author Response · Author response to Decision Letter 0]

4 Jul 2023

The responses to the reviewers are contained in the Response to Reviewer.docx File.

---

## [Editor Report · Decision Letter 1]

18 Jul 2023

An intercomparison of models predicting growth of Antarctic krill (Euphausia superba): the importance of recognizing model specificity

PONE-D-23-06152R1

Dear Dr. Bahlburg,

We’re pleased to inform you that your manuscript has been judged scientifically suitable for publication and will be formally accepted for publication once it meets all outstanding technical requirements.

Kind regards,

Peng Chen, Ph.D.

Academic Editor

PLOS ONE

---

## [Editor Report · Acceptance letter]

20 Jul 2023

PONE-D-23-06152R1 

An intercomparison of models predicting growth of Antarctic krill (*Euphausia superba*): the importance of recognizing model specificity 

Dear Dr. Bahlburg:

I'm pleased to inform you that your manuscript has been deemed suitable for publication in PLOS ONE. Congratulations! Your manuscript is now with our production department. 

Kind regards, 

on behalf of

Dr. Peng Chen 

Academic Editor

PLOS ONE